# N, S Co-Doped Carbons Derived from *Enteromorpha prolifera* by a Molten Salt Approach: Antibiotics Removal Performance and Techno-Economic Analysis

**DOI:** 10.3390/nano12234289

**Published:** 2022-12-02

**Authors:** Mengmeng Zhang, Kexin Huang, Yi Ding, Xinyu Wang, Yingli Gao, Pengfei Li, Yi Zhou, Zheng Guo, Yi Zhang, Dapeng Wu

**Affiliations:** 1School of Business, Henan Normal University, Xinxiang 453007, China; 2Key Laboratory of Green Chemistry Medias and Reactions, Ministry of Education, School of Environment, Henan Normal University, Xinxiang 453007, China; 3College of Textiles, Zhongyuan University of Technology, Zhengzhou 451191, China

**Keywords:** bio-carbon, advanced oxidation, persulfate, sulfamethoxazole, molten salt method

## Abstract

N, S co-doped bio-carbons with a hierarchical porous structure and high surface area were prepared using a molten salt method and by adopting *Entermorpha prolifera* (EP) as a precursor. The structure and composition of the bio-carbons could be manipulated by the salt types adopted in the molten salt assisted pyrolysis. When the carbons were used as an activating agent for peroxydisulfate (PDS) in SMX degradation in the advanced oxidation process (AOP), the removal performance in the case of KCl derived bio-carbon (EPB-K) was significantly enhanced compared with that derived from NaCl (EPB-Na). In addition, the optimized EPB-K also demonstrated a high removal rate of 99.6% in the system that used local running water in the background, which proved its excellent application potential in real water treatment. The degradation mechanism study indicated that the N, S doping sites could enhance the surface affinity with the PDS, which could then facilitate ^1^O_2_ generation and the oxidation of the SMX. Moreover, a detailed techno-economic assessment suggested that the price of the salt reaction medium was of great significance as it influenced the cost of the bio-carbons. In addition, although the cost of EPB-K was higher (USD 2.34 kg^−1^) compared with that of EPB-Na (USD 1.72 kg^−1^), it was still economically competitive with the commercial active carbons for AOP water treatment.

## 1. Introduction

Sulfamethoxazole (SMX) has played a great role in the treatment and prevention of bacterial and fungal infections in clinical health care and in the treatment of animal infections; however, due to its high thermal stability, it was difficult to degrade SMX naturally. This is because it was discharged into the water environment, which could breed drug-resistant bacteria, thus reducing its capability in disease treatment and resulting in potential harm to the ecosystem. At present, traditional water treatment based on biological processes is not effective for removing persistent organic pollutants such as SMX from water; therefore, advanced oxidation processes (AOPs) were developed for the treatment of persistent antibiotics [1].

The AOPs that were based on persulfate (SO_4_^•−^) and hydroxyl (HO·) were effective technologies that removed persistent antibiotic pollutants [2]. Compared with ^•^OH, SO_4_^•−^ has greater oxidization potential, it is environmentally resistant, and it has a prolonged half-life time, which thus endows persulfate-based AOPs with a promising water treatment strategy. Among the persulfates, PDS (peroxydisulfate) is used more widely than PMS (peroxymonosulfate) due to its higher stability and lower price [3]; however, considering that PDS hardly reacts with persistent pollutants, light or heat irradiation, catalytic materials, or activating ions are commonly used to activate the PDS molecules for the production of active oxygen content species (ROSs). These include sulfate radicals, hydroxyl radicals, singlet oxygen, and so on. [4]. Among these PDS activating strategies, carbon materials, with abundant surface defects and edges, were considered as low cost, effective, and environmentally benign catalytic mediums to initiate PDS activation [5].

Biomasses, especially biowastes that are automatically generated from the forestry, agricultural, and food industries, were regarded as abundant and easily available bio-resources that yield low-cost carbon materials for water treatment [6,7,8,9,10,11,12]. As for AOPs, wood chips and ammonium ferric citrate were adopted as precursors to prepare bio-carbons through a multi-step pyrolysis, which had the potential to effectively remove ~99% of SMX in the water system [13]. Peanut shells were modified with iron salt to prepare magnetic bio-carbons with a rich pore structure and high specific surface area; this gave an excellent performance in terms of sulfapyridine and ciprofloxacin degradation [14]. In addition, sugarcane waste [15] and poplar wood powder [16] were also employed as precursors to yield bio-carbons; therefore, they could serve as high performance catalysts to activate PDS to effectively remove bisphenol A.

*Enteromorpha prolifera* (EP) is a marine alga which can survive a wide range of temperature and salinity fluctuations [17]. The massive reproduction of EP has caused a great deal of damage in the ecosystem and to the economies of the fishing and tourism industries. EP biomass is rich in carbon, nitrogen, and iron, which have been employed as precursors to prepare carbon materials for applications in energy storage, heavy metal adsorption, and organic pollutant degradation [18,19,20,21]. For example, EP was adopted to prepare Fe/N co-doped carbon using in situ pyrolysis in a N_2_ atmosphere for the degradation of acetaminophen [20]. EP biomass was also used as precursor to obtain a high performance from Fe_3_C/C composites in a Fenton reaction to degrade methylene blue [21]. Although EP could be employed as an abundant and low-cost biowaste for the preparation of high-performance carbon materials, few studies have prepared carbonous catalysts based on EP for AOPs [22,23,24].

The molten salt method is an effective strategy that yields carbon materials by adopting a low melting point inorganic salt as the protecting medium instead of commonly used inert atmospheres; this means that it is a low cost and energy saving method that prepares carbon materials for different applications [25,26,27,28,29,30]. Based on previous studies, the etching effect of molten salt and the oxygen enriched reaction medium could effectively introduce a hierarchical porous structure as well as rich surface defects, which are favorable for persulfate activation [31,32,33,34]; however, to the best of our knowledge, few studies have devoted themselves to the preparation of bio-carbons, based on the molten salt method, in order to obtain AOPs and study its economic feasibility by systematic tech-economic analysis.

In this study, EP derived bio-carbons were prepared using a facile molten salt method and the effects of a molten salt medium (NaCl and KCl) on the structure and composition of bio-carbons were investigated. When the prepared N, S co-doped porous bio-carbon is adopted in PDS activation for SMX degradation, the bio-carbon that was derived from KCl (EPB-K) shows a higher catalytic removal rate than the bio-carbon derived from NaCl (EPB-Na). The EPB-K also demonstrates good resistance in a varied pH range and it demonstrates great potential in actual water treatment due to the ^1^O_2_ mediated mechanism. Techno-economic analysis indicated that the cost of the molten salt system comprises the greatest portion of carbon price. Although the cost of EPB-K is higher compared with EPB-Na, it is economically competitive with commercial active carbon for AOP water treatment.

## 2. Experiment Section

### 2.1. Preparation of Bio-Carbons

The EP biomass was obtained from Tsingtao, Shandong Province of China, and it was washed with double distilled water and dried at 60 °C, in an oven, overnight. Then, the completely dried EP biomass was ground and sieved with a 100 mesh for later use. The chemical reagents of KCl (AR > 99.5%), NaCl (AR > 99.5%), Pb(NO_3_)_2_ (AR > 99.0%) and Cu(NO_3_)_2_·3H_2_O(AR > 99.0%) were purchased from Sinopharm Chemical Reagent Co., Ltd. (Shanghai, China).

The bio-carbon was prepared using a typical molten salt method wherein 7.0 g of EP powder was mixed with 15.0 g of salt (NaCl or KCl). The mixture was transferred into a porcelain crucible and covered with a layer of salt (1.0 g) to seal the reaction system. Afterwards, the crucible was covered and transferred to a muffle furnace. The temperature was heated to 700 °C at a rate of 10 °C min^−1^, and then it was maintained for 1 h. After it had cooled to room temperature, the yielded bio-carbon was rinsed repeatedly with double distilled water and filtered to remove the salt. The filtered product was also soaked in 1 mol L^−1^ diluted hydrochloric acid for 12 h and filtered and rinsed repeatedly until neutral. The final product was dried to obtain the bio-carbons, which were labeled as EBC-Na and EBC-K, respectively.

The EBC-K was also prepared under different temperatures of 500, 600, 700, and 800 °C with similar pyrolysis profiles in the KCl molten salt medium. The final products were denoted as EBC-500, EBC-600, EBC-700, and EBC-800.

### 2.2. Material Characterization

The morphology and structure were characterized by field emission scanning electron microscopy (SEM, SU8010, Hitachi, Tokyo, Japan) and transmission electron microscopy equipped with energy dispersive spectroscopy (TEM, TF20, JEOL 2100F, Akishima, Japan). The crystalline structure of the sample was determined using X-ray powder diffraction (XRD, Rigaku Dmax-2000, Rigaku, Tokyo, Japan). The graphitization of the bio-carbons was ascertained using Raman spectroscopy (LABRAMHR EVO, HORIBA France SAS, Longjumeau, France), and the surface functional groups and element compositions of EBCs were analyzed using infrared spectrometry (FTIR, Nexus 470, GMI Inc., Lebanon, OH, USA) and X-ray diffraction (XPS, ESCALAB 250Xi, Thermo Fisher Scientific, Waltham, MA, USA). The surface area and porous structure were measured by N_2_ adsorption–desorption analyzer (BET, ASAP2020, Micromeritics Inc., Norcross, GA, USA). 

### 2.3. Advanced Oxidation Performances

Moreover, 0.05 g of the prepared bio-carbon was added into 100 mL and 50 mg L^−1^ SMX solutions in 250 mL conical flasks, respectively, and they were placed on a shaking table with a rotation speed of 180 r min^−1^. In addition, 2 mL of the reaction solution was sampled at 5, 10, 30 and 60 min, respectively, and the solution was filtered with a 0.22 μm membrane to remove the catalysts. The solution was then measured using an UV spectrophotometer (260 nm) in order to monitor the concentration variation; this is related to the adsorption performances of the bio-carbons. After the adsorption equilibrium, 10 mL of the 50 mmol L^−1^ PDS solution was added into the reaction system to start the AOP degradation. Then, 2 mL of the solution was withdrawn at 65, 70, 90, 120, and 180 min. Before the UV spectrophotometer measurement, the sample was filtered with a 0.22 μm membrane and 100 μL methanol was added to quench the reaction. All the tests were averaged using three control trials.

In addition, the performances of the optimized EBC-K were also tested using different initial pH levels, catalyst dosages, and SMX concentrations. In order to further explore potential applications in real water treatment scenarios, SMX was also added to the running water in the background to test the degradation rate of EBC-K. Please find the experimental details in the Appendix A.

## 3. Result and Discussion

### 3.1. Structure and Composition Characterization

Figure 1a,b shows how the Enteromorpha prolifera (EP) invaded the seashore as well as the collected EP biomass in the salvage yard at Tsingtao City, Shandong Province of China. As depicted in Figure 1c, the fresh EP biomass exhibits a smooth surface without porous tissue structures; however, after the molten salt treatment (KCl), the prepared bio-carbon (EBC-K) exhibited sheet-like patterns with well-developed porous structures (Figure 1d). The SEM image with high magnification, shown in Figure 1e, shows the rich porous structures with diameters of several tens of nanometers; this was possibly caused by the etching effects of both the KCl and the oxygen which is penetrated through the molten salt [32]. Figure 1f depicts how the EBC-Na that was derived from NaCl molten salt demonstrates sheet-like patterns with porous structures that have diameters of ~100 nm. In addition, based on the TEM images displayed in Figure 1g,h, the EBC-K possesses regularly distributed mesoporous structures with diameters of 2–5 nm. The corresponding selected area, as shown in the electron diffraction (SAED) image, discloses two sets of indistinct diffraction rings that are attributed to the carbon lattice; this indicates that the EBC-K has a medium graphite structure. As indicated by the HAADF-STEM and EDS mappings of EBC-K (Figure 1i), C, O, N and S are found homogeneously distributed on the surface of the bio-char. As no N or S contented regents were adopted in the synthesis, the rich N and S originated from the intrinsic composition of the EP biomass. Moreover, slight amounts of Si and K could be also found embedded in the bio-carbon; this may also be ascertained from the intrinsic composition of the biomass (Appendix A).

Appendix A depicts the XRD characterization of the bio-carbons, which possess typical diffraction peaks that can be ascribed to the (002) and (100) planes of carbon, respectively [31,32]. In addition, several sharp peaks belonging to the insolvable mineral products could be also detected; this indicates that the mineral contents in EP could be converted into insoluble ceramics during the high temperature treatment. As shown in Figure 2a, the FTIR peak centered at ~1049 cm^−1^ can be ascribed to the COOH or CHO bending vibration, whereas the characteristic peak centered at ~1622 cm^−1^ occurs due to the C=O stretching vibration [33,34]. The peak intensities of EBC-K correspond with these oxygen content function groups, and they are slightly increased compared with EBC-Na, thus indicating that the population in the surface function group is greatly enhanced. As shown in Figure 2b, the Raman spectra demonstrate two characteristic peaks that correspond with the D band (~1300 cm^−1^); these originated from defects and from the G band (~1600 cm^−1^) in the in-plane *sp^2^* vibrational mode of the carbon. The normalized Raman curves indicate that the I_D_/I_G_ ratio could be measured as 1.01 and 0.97 for EBC-K and EBC-Na; this indicates that fewer surface defects are formed in the EBC-Na [35,36,37,38]. This phenomenon may possibly be ascribed to the fact that the KCl with a lower melting point could react better with the carbon skeleton to create abundant surface defect sites on the carbon skeleton. The N_2_ adsorption–desorption isotherms and the pore size distribution of EBC-K and EBC-Na are displayed in Figure 2c,d. The curves both demonstrated typical IV curves, which indicate the co-existence of micro- and meso-porous structures [39]. In addition, the hysteresis loops could be observed in the isotherms of both of the bio-carbons, thus suggesting the existence of meso-porosity which possibly results from the tissue structure of the EP; however, the N_2_ adsorption of the EBC-K was much enhanced compared with that of the EBC-Na, which gives rise to the greater BET surface area of 356.8 m^2^ g^−1^, compared with that of EBC-Na (257.4 m^2^ g^−1^). Furthermore, the N_2_ adsorption that occurred with relative pressure from 0–0.2 and 0.4–0.8 were related to the microporous and mesoporous structures, respectively. It is obvious that the two bio-carbons exhibited a similar microporous structure, but the EBC-K was dominated by mesopores [40]. As shown in Figure 2d, EBC-K possesses a higher pore size distribution in both the micro- and meso-porous range. In addition, the average pore size of the EBC-K was greatly increased to 2.9 nm compared with that of the EBC-Na (2.3 nm); this results in an increased number of increased active sites for both PDS activation and SMX adsorption. In addition, EBC-K also demonstrates a higher pore volume of 0.258 m^3^ g^−1^ compared with that of EBC-Na (0.151 m^3^ g^−1^); this indicates that EBC-K exhibits higher mass diffusion dynamics during SMX degradation.

The compositions of the bio-carbons were also characterized by X-ray photoelectron spectroscopy (XPS). As shown in Appendix A, the survey spectra of the two samples exhibit prominent C, O, N and S signals, thus indicating that the N, S elements originated from the EP biomass. Moreover, EBC-K possesses relatively high N (4.85%), S (2.35%), and O (24.07%) contents compared with that of EBC-Na (N (4.77%), S (1.96%) and O (19.09%)), thus suggesting that the KCl molten salt system could better preserve the heteroatoms in the bio-carbon; this is in accordance with the Raman spectra. Further analysis of the high-resolution peaks is depicted in Figure 3a–d. The C1s signals could be deconvolved into four peaks, which correspond with C-C/C=C, C-O, C=O and COOH, respectively (Figure 3a) [41,42]. As shown in Figure 3b and Table 1, the greater O content of EBC-K is largely derived from the enhanced COOH on the surface; this is considered to be a reliable adoption site for SMX. In addition, the high-resolution N spectra in Figure 3c could be separated into four peaks belonging to pyridinic N, pyrrolic N, graphitic N, and oxidized N, respectively [43,44]. EBC-K shows a much higher pyridinic N content (74.59%) than that of the EBC-Na (68.64%), which means that it could serve as an active sit for PDS activation. On the other hand, it was reported that the S doping sites could give rise to the asymmetric electron spin and charge polarization on the carbon surface, which could accelerate the adsorption of PDS to facilitate the AOP process. As depicted in Figure 3d and Table 1, the high-resolution S 2p spectra of EBC-K indicates that the two peaks centered at ~163 eV and ~164 eV are assigned to S 2p_3/2_ and S 2p_1/2_ (thiophene-S), respectively, and the other two that are centered at ~167 eV and 168 eV correspond with the oxidized S (-SO_x_) [45,46]. Compared with EBC-Na, the EBC-K with an enhanced portion of lower valance thiophene-S (42.21%) and oxidized S (24.07%) are expected to better activate the PDS for the AOPs.

### 3.2. AOP Performances

The PDS activation performances of the bio-carbons were explored by using SMX as the simulating pollutant. In order to better illustrate the adsorption and PDS degradation, the measurements were divided into two phases. After the adsorption reaches equilibrium after 60 min, the PDS was subsequently added into the system to start the AOP. As shown in Figure 4a, EBC-K shows a much higher adsorption capability as well as degradation process compared with that of EBC-Na. The overall removal rate could achieve up to 92.7% a significant improvement compared with the removal rate of EBC-Na (65.9%); therefore, the detailed optimization of the preparation parameters was carried out in the KCl system. Based on Figure 4b, it was found that without a catalyst, the SMX experiences no obvious degradation with the addition of PDS, thus indicating that the self-activation of PDS is rather low; however, for the degradation system without PDS, the removal of SMX is solely caused by the adsorption process, and the high adsorption capability of EBC-K (~50%) could facilitate the later degradation process. As demonstrated in Figure 4c, the EBC-700 that was prepared under 700 °C possessed the best performance. The higher pyrolysis temperature inevitably leads to the evaporation of the slat medium, which leads to lower adsorption and catalytic performances. 

In addition, the pH condition is of great significance to the SMX degradation rate as it affects the surface adsorption and reactive species formation process. As shown in Figure 4d, although the high pH leads to much lower adsorption performances, an overall degradation rate of up to ~75% can nevertheless be achieved. Moreover, Figure 4e indicates that SMX degradation could be further increased to ~100% by increasing the catalytic dosages to 1.00 g. Figure 4f shows that even with lower SMX concentrations, the adsorption and degradation rates experience obvious increases as expected, and the maximum removal rate could reach up to 99.7% when the concentration is reduced to 20 mg L^−1^.

### 3.3. Degradation Mechanism

In order to determine the reactive species generated by the PDS activation, systematical radical quenching experiments were performed. As shown in Figure 5a, MeOH was adopted to salvage the SO_4_^•−^ and ^•^OH active species. The degradation performance of EBC-K was barely influenced. To further confirm the role of ^•^OH, TBA was introduced to quench the ^•^OH without affecting the SO_4_^•−^ (Figure 5b). Although the overall level of degradation was slightly reduced, this reduction possibly resulted from the surface adsorption sites which were partially occupied by the TBA; this finally gave rise to the lower adsorption accommodation for SMX [47,48,49]. This evidence indicates that SO_4_^•−^ and ^•^OH were not the dominating reactive species in SMX degradation. In addition, L-histidine was adopted to quench ^1^O_2_ during the degradation process and the results were shown in Figure 5c. After increasing the L-histidine amounts, SMX degradation drops to 48.3%. In order to further confirm the generation of ^1^O_2_, FFA was adopted to trap ^1^O_2_ (Figure 5d). After increasing the FFA concentration, the SMX degradation rate decreased to 50.4%, thus confirming the fact that ^1^O_2_ serves as the dominating reactive species in SMX degradation [50]. To better understand the role of EBC-K in real AOP scenarios, running water from the local water supply company was used in the background in order to simulate the SMX pollution. As shown in Appendix A, although the intrinsic organic compounds and intervening ions inhibit the degradation process, it was found that over 85% of the degradation rate could be achieved (50 mg L^−1^), which is relatively lower than the rate that can be achieved with pure water. If the SMX concentration is further reduced to 20 mg L^−1^, the removal rate could reach to 99.6%, which is similar to the rate achieved with the pure water system.

### 3.4. Tech-Economic Assessment

The annual explosive growth of EP between spring and summer brings forth a tremendous hazard to the local ecosystem as well as the fishing and tourism industries in Tsingtao City, Shandong Province of China; therefore, the local government offers millions of dollars in subsidies to clean them from the sea, and millions of tons of EP biomass are collected and stored. Only a small portion can be utilized as fertilizer or biomass for bio-oil production; therefore, it is imperative to develop an economic treatment for EP biomass in order to minimize the potential hazards. Based on these considerations, the bio-carbon factory will be established 100 km away from the EP storage yard in Tsingtao City. The cost of manufacturing (COM) is expected to anticipate the economic cost of bio-carbon produced by EP in the process of industrial manufacturing. Based on previous reports, COM is composed of five main parts: the fixed capital investment (FCI), costs of labors (C_OL_), raw materials (C_RM_), waste treatment (C_WT_) and utilities (C_UT_). Moreover, the COM of bio-carbon can therefore be calculated with the following equation, and the detailed calculation is presented with Appendix A [51].
COM = 0.230FCI + 2.73C_OL_ + 1.23(C_RM_ + C_WT_ + C_UT_)

As shown in Figure 6a, the production procedure of bio-carbon includes three steps: EP transportation, mixing process of biomass and salt, and the high temperature pyrolysis process. The industrial furnace, which has the capacity of ~3000 L, a temperature of 800 °C, and a maximum power of 150 kW, is necessary. Moreover, a mixing machine with a capacity of ~3000 L and a maximum power of 30 kW is equipped with the furnace. Four sets of furnaces and mixing machines will be installed in the factory. The price of a furnace is USD 3500 and a mixing machine is USD 3000. The FCI could be estimated as costing USD 26,000. The average wage when working in the manufacturing industry in Shandong Province is USD 3.57 h^−1^ (National Bureau of Statistics of China, 2021). Factories tend to operate 330 days a year with 24 h shifts which require four employees per shift. Based on this, the total annual C_OL_ could be calculated as USD 226,195.2. The C_RM_ includes the EP biomass and the molten salt, which are directly used for bio-carbon production. As the local government will subsidize the cost of EP collection and storage, the cost of the EP could be estimated by using the transportation fee from the storage yard to the factory. The local transportation price is USD 0.0917 per ton-kilometer, and the price of the EP is calculated as USD 9.17 ton^−1^. The factory processes 5280 tons of EP every year, thus, the annual cost of biomass is USD 48,417.6. During the molten salt process, the weight ratio of salt and biomass is 1:1, and 50% of the salt can be recycled. In addition, the price of KCl and NaCl is USD 260 and USD 80 ton^−1^, respectively; therefore, the costs of KCl and NaCl are estimated as USD 686,400 and USD 211,200 every year, respectively; therefore, the total annual C_RM_ is USD 734,817.6 and USD 259,617.6. As the factory recycles approximately half of the water and salt in the production process, the C_WT_ can be ignored. The C_UT_ refers to the electricity cost consumed by the furnaces and the mixing machines. The industrial and commercial electricity price is USD 0.099 h^−1^ in Shandong Province; therefore, the total annual C_UT_ is USD 564,357.6. The carbon yield in the production process is estimated as ~18%, and the COMs of the bio-carbon using KCl and NaCl in the production process are USD 2.34 kg^−1^ and USD 1.72 kg^−1^, respectively; these prices can compete with the commercially active carbons. As shown in Figure 6b,c, the sensitivity analysis of COM remains stable as the market price fluctuates. In addition, as depicted in Figure 6d, a 10% fluctuation of C_RM_ leads to about a 4.27% COM variation, indicating that, in the case of KCl, the C_RM_ is the primary contributor to the COM. On the other hand, Figure 6e further suggests that when the NaCl is used as the reaction medium, a 10% fluctuation in C_UT_ leads to 4.07% COM variation, thus indicating that the C_UT_ is the main contributor of the COM in the case of NaCl. The tech-economic analysis concludes that the COM is significantly impacted by the cost of salt and the electricity; therefore, the molten salt method with low-price salt, as well as with a low temperature operation process, should be developed to further reduce the COM and the emissions from bio-carbon production.

## 4. Conclusions

*Entermorpha prolifera*, a widely distributed harmful algae, was employed as precursor for the first time in order to prepare N, S co-doped porous bio-carbon with a hierarchical porous structure and high surface area (356.8 m^2^ g^−1^), and KCl was adopted as the molten salt medium. It was found that the KCl with a low melting point was more reactive, thus achieving a high surface area and introducing rich surface function groups; this gives rise to high PDS adsorption and rapid ^1^O_2_ generation levels. Therefore, the optimized EBC-K demonstrated a high SMX removal rate in both the experimental and applicational scenarios. Moreover, the techno-economic assessment indicated although the cost of EPB-K is higher (USD 2.34 kg^−1^) compared with the EPB-Na (USD 1.72 kg^−1^), EPB-K is still economically competitive with the commercial carbon. The cost sensibility analysis indicates that the cost of the salt medium and the electricity are the primary contributors to the COM. This work demonstrates that the EP could be employed as an abundant biowaste that produces bio-carbons with high performances, and the molten salt strategy could be further optimized by adopting low-cost salt to reduce the COM.

## Figures and Tables

**Figure 1 nanomaterials-12-04289-f001:**
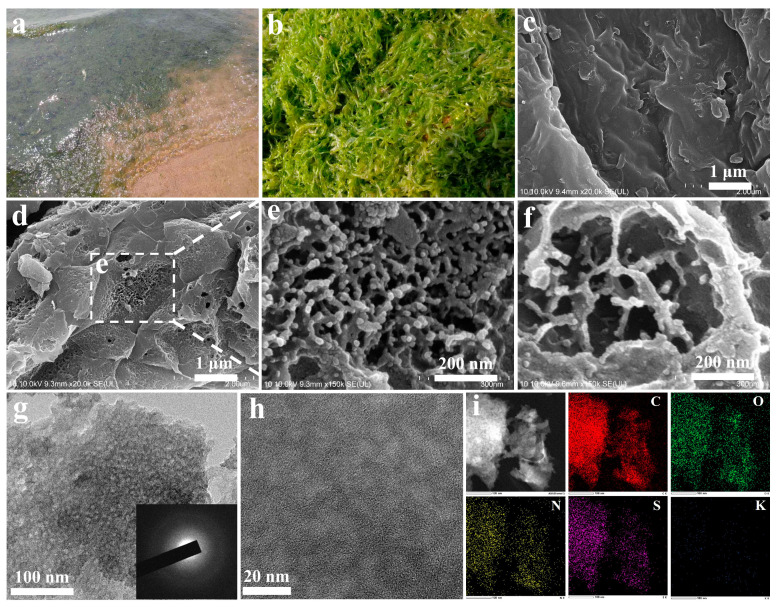
The photographs of (**a**) an EP invaded seashore at Tsingtao City, Shandong Province of China, and (**b**) the fresh EP biomass. The SEM images of (**c**) fresh EP, (**d**,**e**) EBC-K, and (**f**) EBC-Na. The TEM images with (**g**) low and (**h**) high magnification of EBC-K; the inset is the SAED image. (**i**) HAADF-STEM image and EDS element distribution of EBC-K (red: carbon, green: oxygen, yellow: nitrogen, magenta: sulfur and blue: potassium).

**Figure 2 nanomaterials-12-04289-f002:**
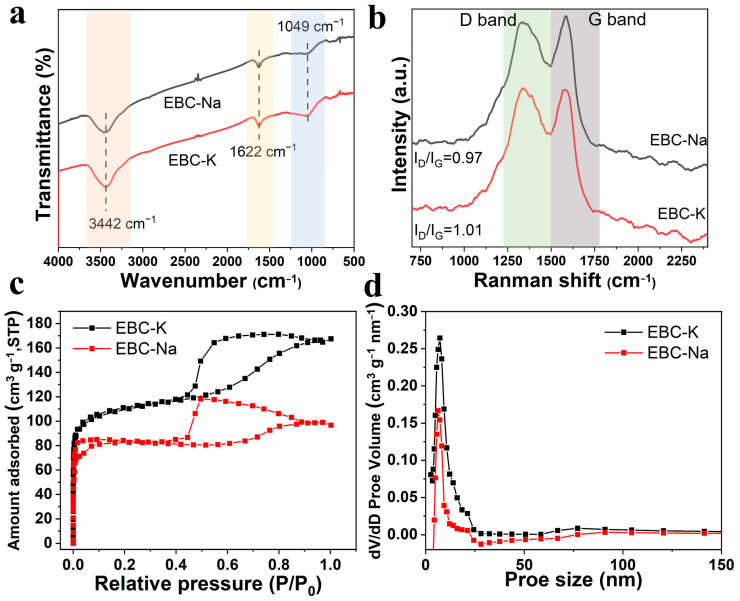
(**a**) FTIR and the (**b**) Raman curves of the EBC-Na and EBC-K. (**c**) N_2_ adsorption/desorption isotherms and (**d**) pore size distribution of EBC-Na and EBC-K.

**Figure 3 nanomaterials-12-04289-f003:**
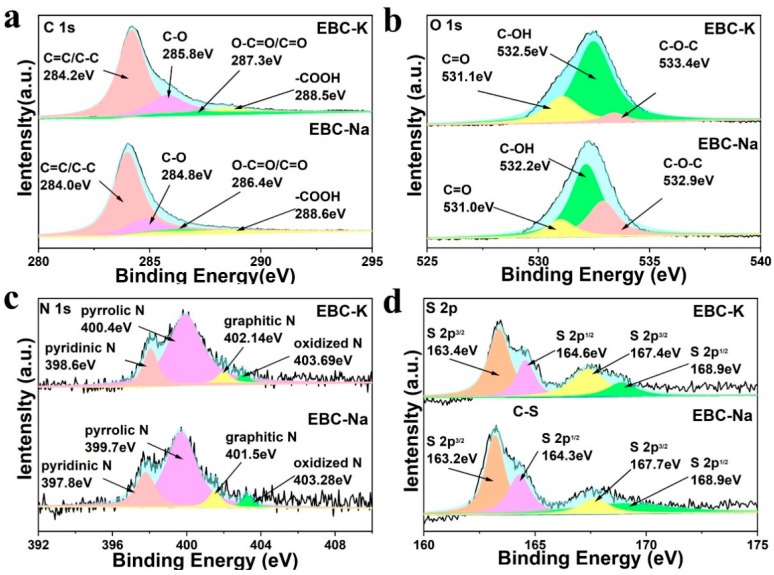
High resolution XPS spectrum of EBC-Na and EBC-K, (**a**) C 1s, (**b**) O 1s, (**c**) N 1s, and (**d**) S 2p.

**Figure 4 nanomaterials-12-04289-f004:**
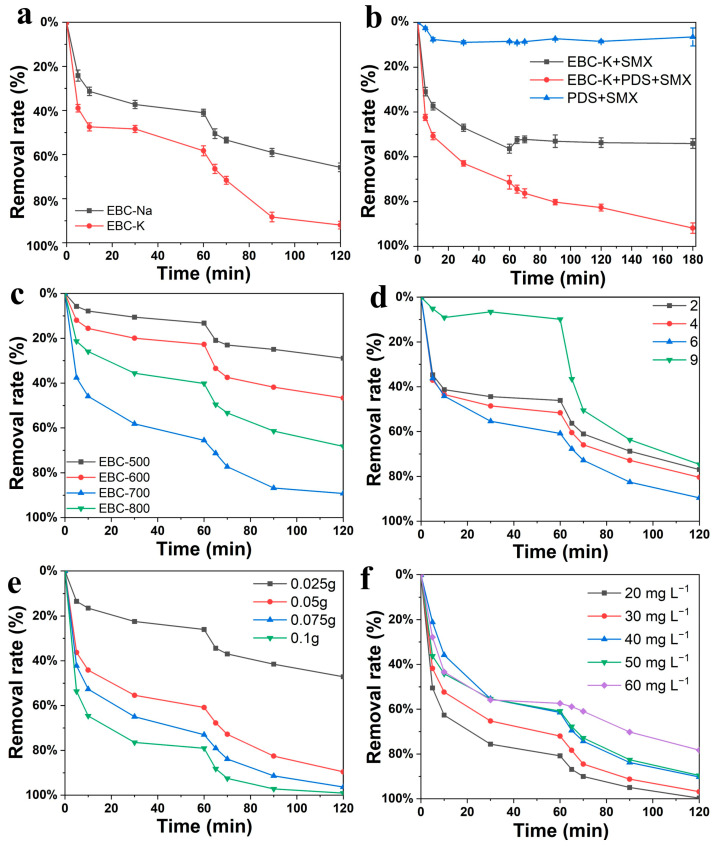
(**a**) The SMX degradation performances of EBC-K and EBC-Na, (**b**) the degradation under different degradation conditions, the detail degradation performances of (**c**) EBC-Ks prepared under varied pyrolysis temperatures, and the optimized EBC-K measured at varied (**d**) initial pH levels, (**e**) bio-carbon dosages, and (**f**) SMX concentrations.

**Figure 5 nanomaterials-12-04289-f005:**
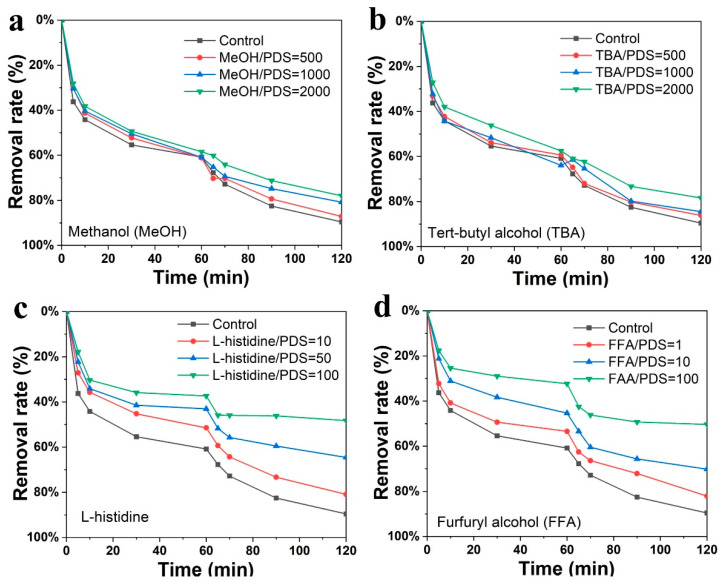
SMX degradation performances of EBC-K measured with different scavengers: (**a**) methanol (MeOH), (**b**) tert-butyl alcohol (TBA), (**c**) L-histidine, and (**d**) furfuryl alcohol (FFA). (Control, in the inset legends, represents the test without a scavenger, and the values represent the molar ratio between the scavenger and the PDS.).

**Figure 6 nanomaterials-12-04289-f006:**
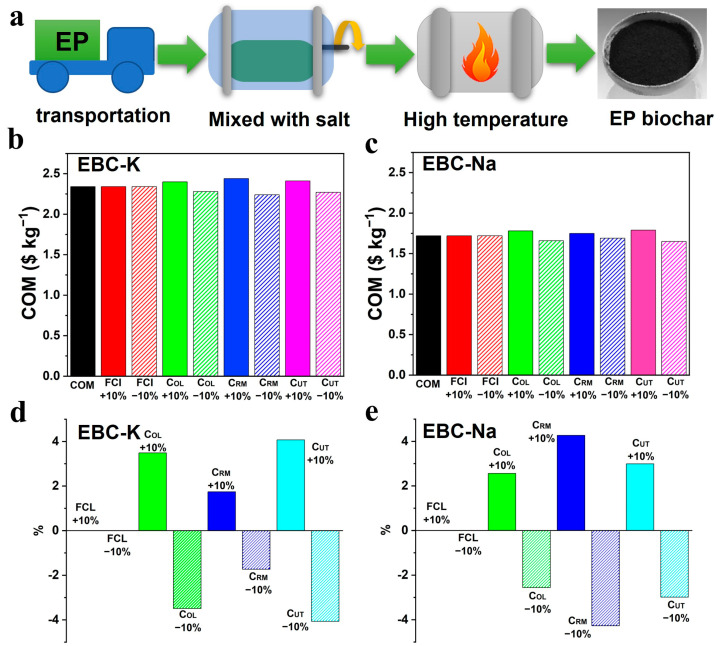
(**a**) The manufacturing processes of the bio-carbons derived from EP using the molten salt method; the COM of (**b**) EBC-K and (**c**) EBC-Na; the sensitivity analysis of the COM variation of (**d**) EBC-K and (**e**) EBC-Na upon the cost fluctuation with 10% increase or decrease.

**Table 1 nanomaterials-12-04289-t001:** The fitting results of the high resolution XPS spectra of C, O, N, and S for EBC-K and EBC-Na.

Bio-Carbons	C Distribution (%, atm)				O Distribution (%, atm)			
	C=C /C-C	C-O	O-C=O /C=O	COOH	C=O	C-OH		C-O-C
**EBC-K**	75.29	19.75	0.79	4.17	22.37	72.39		5.24
**EBC-Na**	76.95	18.34	3.42	1.29	13.13	59.39		27.48
	**N Distribution (%, atm)**				**S Distribution (%, atm)**			
	**Pyridinic**	**Pyrrolic**	**Graphitic**	**Oxidized**	**C-S**		**Oxidized Sulfur**	
					**S 2p_3/2_**	**S 2p_1/2_**	**S 2p_3/2_**	**S 2p_1/2_**
**EBC-K**	17.35	74.59	4.78	3.28	42.21	19.27	27.03	11.49
**EBC-Na**	20.76	68.64	6.36	4.24	39.32	23.28	11.32	26.08

## Data Availability

The detailed data in the study are available from the corresponding authors by request. (Dapeng Wu and Pengfei Li).

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
