# Peer review of "N, S Co-Doped Carbons Derived from Enteromorpha prolifera by a Molten Salt Approach: Antibiotics Removal Performance and Techno-Economic Analysis"

_nanomaterials, 2022, doi:10.3390/nano12234289_

Round 1

Reviewer 1 Report

Review of the Manuscript No Nanomaterials-2050507

1. Authors should improve English text and rewrite the Abstract, Introduction, Results and Conclusions.  

Abstract

“As employed for the peroxydisulfate (PDS) activating to degrade SMX in advanced oxidation process (AOP), the removal performance of the KCl derived bio-carbon (EPB-K) was greatly enhanced compared with the bio-carbon derived from NaCl (EPB-Na).”

should be substituted with

“It was established that the removal performance in the case of KCl derived bio-carbon (EPB-K) was significantly enhanced, compared with the bio-carbon derived from NaCl (EPB-Na), when the carbons were used as activating agent for peroxydisulfate (PDS) in SMX degradation in advanced oxidation process (AOP).”

2. Introduction, page 1, row 34-35

“At present, traditional water treatment based on biological processes has short comes in removing persistent organic pollutants such as SMX from the water bodies.”

Should be replaced by

“At present, traditional water treatment based on biological processes is not effective for removal persistent organic pollutants such as SMX from the waters.”

3. Page 11, Row 332

“Conclusion” should be corrected to “Conclusions”

4. Please correct the references according to all journal requirements. Add DOIs of the references.

Author Response

Nanomaterials-2050507

Detailed responses to the reviewers

Reviewer 1

  1. Authors should improve English text and rewrite the Abstract, Introduction, Results and Conclusions.

Abstract “As employed for the peroxydisulfate (PDS) activating to degrade SMX in advanced oxidation process (AOP), the removal performance of the KCl derived bio-carbon (EPB-K) was greatly enhanced compared with the bio-carbon derived from NaCl (EPB-Na).” should be substituted with “It was established that the removal performance in the case of KCl derived bio-carbon (EPB-K) was significantly enhanced, compared with the bio-carbon derived from NaCl (EPB-Na), when the carbons were used as activating agent for peroxydisulfate (PDS) in SMX degradation in advanced oxidation process (AOP).”

Reply: Thanks for your kind comments, we have revised this section accordingly. In addition, based on your comments on our language, we have thoroughly revised the language of the manuscript. Moreover, we have also sent our paper to a colleague who has been working in USA for about 10 years and published over 50 high impact papers in material science (such as Adv. Mater, Agnew, JACS and so on). With his generous help, the manuscript was proofed to remove the grammar errors concerning about the tension, plural form, spelling as well as the sentence structure, which substantially enhance the language quality of the manuscript.

Please find these corrections in the revised manuscript and the sections received correction were marked in red.

  1. Introduction, page 1, row 34-35

“At present, traditional water treatment based on biological processes has short comes in removing persistent organic pollutants such as SMX from the water bodies.”

Should be replaced by

“At present, traditional water treatment based on biological processes is not effective for removal persistent organic pollutants such as SMX from the waters.”

Reply: Thanks a lot for your comments, and we have revised this section based on your suggestion.

  1. Page 11, Row 332

“Conclusion” should be corrected to “Conclusions”

Reply: Thanks for your comments and we have changed this in the revised version.

  1. Please correct the references according to all journal requirements. Add DOIs of the references.

Reply: Based on your comments, we have strictly revised the format of the references and the DOIs were added at the end of each reference.

Reviewer 2 Report

Dear Authors

The manuscript is focused on the N, S co-doped bio-carbons with hierarchical porous structure and high surface area that were prepared from Entermorpha prolifera (EP) as precursor through a molten-salt method.

The following suggestion and comments should be taken:

1. The overall English needs to be improved. Please seek guidance from a native English speaker if possible ("the" "a", commas, plural form and others could be corrected).

2. The introduction section needs enhancement few sentences about different activated carbons, their modifications with N, S and potential applications. Please cite:

(1) Materials 2021, 14(14), 3996; https://doi.org/10.3390/ma14143996

(2) Nanomaterials 2022, 12(18), 3156; https://doi.org/10.3390/nano12183156

(3) Nanomaterials 2021, 11(9), 2217; https://doi.org/10.3390/nano11092217

3. Could the authors include the standard deviation of the used methods?

4. Is SEM/EDS a good method to analyze elements such as C or O? Please explain.

5. Figure 1. Please correct this image for better quality (the inscriptions).

6. Figure 5 please correct this image (description - control/500/1000/2000) for better quality.

7. Why author choose activated carbons for the study? Please explain in the comments.

8. EDX and XPS methods should have a different ratio of C:O in these two methods, what do the authors say about that?

9. I feel this paper has not given an extensive report on Brunauer-Emmett-Teller (BET) Surface Area Analysis and Barrett-Joyner-Halenda (BJH) Pore Size and Volume Analysis. What kind of pores do the authors have please comment it.

10. Authors are suggested to describe some future plans in conclusions.

Author Response

Nanomaterials-2050507

Detailed responses to the reviewers

Reviewer 2

The manuscript is focused on the N, S co-doped bio-carbons with hierarchical porous structure and high surface area that were prepared from Entermorpha prolifera (EP) as precursor through a molten-salt method.

The following suggestion and comments should be taken:

  1. The overall English needs to be improved. Please seek guidance from a native English speaker if possible ("the" "a", commas, plural form and others could be corrected).

Reply: thanks a lot for your kind comments and we have thoroughly revised the English of this manuscript. Moreover, we have also sent our paper to a colleague who has been working in USA for about 10 years and published over 50 high impact papers in material science (such as Adv. Mater, Agnew, JACS and so on). With his generous help, the manuscript was proofed to remove the grammar errors concerning about the tension, plural form, spelling as well as the sentence structure, which substantially enhance the quality of the manuscript.

Please find the detailed revision in the revised manuscript and the places received corrected was marked in red.

  1. The introduction section needs enhancement few sentences about different activated carbons, their modifications with N, S and potential applications. Please cite:

(1) Materials 2021, 14(14), 3996; https://doi.org/10.3390/ma14143996

(2) Nanomaterials 2022, 12(18), 3156; https://doi.org/10.3390/nano12183156

(3) Nanomaterials 2021, 11(9), 2217; https://doi.org/10.3390/nano11092217

Reply: Thanks for your comments, we have carefully read through these papers which are highly related with our research topic, and these papers are cited in the revised manuscript as references 11, 12 and 40.

  1. Could the authors include the standard deviation of the used methods?

Reply: Thanks for your nice comments, we have mentioned in the experimental section that the data are average from three identical measurements. We have provided the standard deviation of the degradation in Figure 4a and b. Please find them in the revised manuscript.

  1. Is SEM/EDS a good method to analyze elements such as C or O? Please explain.

Reply: Thanks for your comments. As we have stated in the discussion, the element distribution of the sample is measured by the STEM/EDS system. The EDS measurement is performed by the follow steps. Firstly, the high energy electron beam could excite the electron of the elements to the higher energy level. And the pumped electron will relax to the lower energy level to generate energy specified X ray, which could indicate the difference of the element. During this test, the X ray energy is related to the element type and independent with the incident irradiation energy. On the other hand, the detecting depth of the electron beam from the SEM instrument is about 100 nanometers, which could effectively detect the element composition of the materials surface.

Generally speaking, the EDS method is capable to analyze the elements which are heavier than borne. Therefore, it is reasonable to detect the C and O distribution in the samples.

  1. Figure 1. Please correct this image for better quality (the inscriptions).

Reply: Thanks for your suggestion, we have remade the figure 1, to better quality based on the requirement of the journal.

  1. Figure 5 please correct this image (description - control/500/1000/2000) for better quality.

Reply: Thanks for your reminding, we have further explained the parameters in both the figure and the figure captions to better illustrate the details in the degradation tests with different quenchers.

  1. Why author choose activated carbons for the study? Please explain in the comments.

Reply: Thanks for your comment, the activated carbon is a fully commercialized low cost and effective materials for energy storage, catalytic and adsorption applications. Therefore, the active carbon is regard as a benchmark to evaluate the performances of other products. In addition, we have also consulted with many research papers, active carbon is also widely regarded as reference for the tech-economic assessment in the carbon production. Based on these considerations, we used active carbon in our study.

  1. EDX and XPS methods should have a different ratio of C:O in these two methods, what do the authors say about that?

Reply: Thanks for your nice comments, we have to admitted that the two types of methods will definitely give rise to the different ratio of C:O. this deviation is closely related to the detecting depth of the EDX and XPS methods. For the XPS measurement, the detecting depth is limited to several nanometers at the surface of the product, which is better to analyze the surface composition of the product. Meanwhile, the detecting depth of EDX is higher than XPS and could amount to several hundreds of nanometers, which could better reflect the inner composition beneath the surface. Due to the detecting difference of the two instruments, the variation of C:O ratio is reasonable.

  1. I feel this paper has not given an extensive report on Brunauer-Emmett-Teller (BET) Surface Area Analysis and Barrett-Joyner-Halenda (BJH) Pore Size and Volume Analysis. What kind of pores do the authors have please comment it.

Reply: Thanks for pointing out his problem, we have provided with more detailed discussion on the results of the BET surface area analysis and the BJH pore size of the samples. All the parts received corrections was marked in red.

  1. Authors are suggested to describe some future plans in conclusions.

Reply: Thanks a lot for this suggestion, we have involved a brief discussion about the future of this proposed technology in the conclusions of revised manuscript.

Round 2

Reviewer 2 Report

Dear Authors

The authors have addressed all comments and the manuscript can be published as is.